

# Phylomitogenomics elucidates the evolution of symbiosis in Thoracotremata (Decapoda: Cryptochiridae, Pinnotheridae, Varunidae)

Tao Xu[1], Henrique Bravo[1] and Sancia E.T. van der Meij[1,2]

[1] Groningen Institute for Evolutionary Life Sciences, University of Groningen, Groningen, Netherlands
[2] Marine Biodiversity Group, Naturalis Biodiversity Center, Leiden, Netherlands

## ABSTRACT

**Background**. Thoracotremata belong to the large group of ''true'' crabs (infraorder Brachyura), and they exhibit a wide range of physiological and morphological adaptations to living in terrestrial, freshwater and marine habitats. Moreover, the clade comprises various symbiotic taxa (Aphanodactylidae, Cryptochiridae, Pinnotheridae, some Varunidae) that are specialised in living with invertebrate hosts, but the evolutionary history of these symbiotic crabs is still partially unresolved.

**Methods**. Here we assembled and characterised the complete mitochondrial genomes (hereafter mitogenomes) of three gall crab species (Cryptochiridae): *Kroppcarcinus siderastreicola*, *Opecarcinus hypostegus* and *Troglocarcinus corallicola*. A phylogenetic tree of the Thoracotremata was reconstructed using 13 protein-coding genes and two ribosomal RNA genes retrieved from three new gall crab mitogenomes and a further 72 available thoracotreme mitogenomes. Furthermore, we applied a comparative analysis to characterise mitochondrial gene order arrangement, and performed a selection analysis to test for selective pressure of the protein-coding genes in symbiotic Cryptochiridae, Pinnotheridae, and Varunidae (*Asthenognathus inaequipes* and *Tritodynamia horvathi*).

**Results**. The results of the phylogenetic reconstruction confirm the monophyly of Cryptochiridae, which clustered separately from the Pinnotheridae. The latter clustered at the base of the tree with robust branch values. The symbiotic varunids *A. inaequipes* and *T. horvathi* clustered together in a clade with free-living Varunidae species, highlighting that symbiosis in the Thoracotremata evolved independently on multiple occasions. Different gene orders were detected in symbionts and free-living species when compared with the ancestral brachyuran gene order. Lastly, the selective pressure analysis detected two positively selected sites in the *nad6* gene of Cryptochiridae, but the evidence for positive selection in Pinnotheridae and *A. inaequipes* and *T. horvathi* was weak. Adaptive evolution of mitochondrial protein-coding genes is perhaps related to the presumably higher energetic demands of a symbiotic lifestyle.

Corresponding author
Tao Xu, tao.xu@rug.nl

## INTRODUCTION

Brachyuran crabs, one of the most diverse groups of crustaceans, were divided into three (sub) sections based on the position of the gonopore by *Guinot (1978)* and *Guinot (1979)*: Podotremata, Heterotremata, and Thoracotremata. Podotremata have since been shown to be paraphyletic, whereas the reciprocal monophyly of Heterotremata and Thoracotremata is supported (*Tsang et al., 2014*). Thoracotemata consist of five formalised superfamilies (Aphanodactyloidea, Grapsoidea, Ocypodoidea, Pinnotheroidea and Cryptochiroidea) together comprising 21 families (*DecaNet, 2023*), however, *Tsang et al. (2022)* have proposed a new division into seven superfamilies. Thoracotreme crabs inhabit widely different habitats in terrestrial, freshwater and marine environments across the world. Evolutionary adaptations to these different environments could be the result of pressure exerted by the environments themselves and/or by the crabs' innate ability to adapt to new habitats. For example, the majority of Grapsoidea and Ocypodoidea crabs are free-living and can be found in almost all reported habitats for brachyuran crabs (*Tan et al., 2016*; *Wang et al., 2020*), while Pinnotheroidea (pea crabs), Cryptochiroidea (gall crabs) and Aphanodactyloidea live in symbiotic relationships with specific invertebrate hosts, albeit with rare exceptions (*Castro, 2015*). Pea crabs associate with bivalves, gastropods, echinoids, holothurians, polychaetes, other crustaceans, and ascidians (*Castro, 2015*; *Palacios Theil, Cuesta & Felder, 2016*; *Hultgren, Foxx & Palacios Theil, 2022*), gall crabs inhabit dwellings in scleractinian corals (*Fize & Serène, 1957*; *Kropp, 1990*), and aphanodactylid crabs associate with tube building polychaete worms (*Poore & Ahyong, 2023*). Symbionts within Thoracotremata are also found in Varunidae (superfamily Grapsoidea): *e.g.*, *Sestrostoma balssi* (Shen, 1932), *S. depressum* (Sakai, 1965), and *S. toriumii* (Takeda, 1974) reside in the tubes of callianassids, upogebiids, echiurans, and polychaetes (*Castro, 2015*); *Asthenognathus inaequipes* Stimpson, 1858, inhabits holothurians (*Lee, Lee & Ko, 2010*); and *Tritodynamia horvathi* Nobili, 1905, associates with polychaetes (*Sakai, 1976*; *Otani, Takahashi & Matsuura, 1996*).

The monophyly of the Thoracotremata has been confirmed by various studies (*Von Sternberg & Cumberlidge, 2001*; *Tsang et al., 2014*; *Wang et al., 2020*), but the monophyly of the superfamilies within the Thoracotremata has long been debated and discussed (*Schubart et al., 2006*; *Tsang et al., 2014*; *Tsang et al., 2018*; *Tsang et al., 2022*; *Van der Meij & Schubart, 2014*; *Chen et al., 2018a*; *Ma et al., 2019*; *Sun et al., 2022*). Aphanodactylidae and Pinnotheridae display similar overall macromorphology, but are distantly related. *Poore & Ahyong (2023)* elevated Aphanodactylidae, previously considered a pinnotheroid, to superfamily level and Pinnotheroidea now consists of the families Pinnotheridae, Parapinnixidae and Tetriasidae (*Tsang & Naruse, 2023*). Whilst the Cryptochiridae are classified in their own superfamily, the Cryptochiroidea, *Wetzer, Martin & Boyce (2009)* questioned this based on molecular data (16s rRNA) and proposed Cryptochiridae to be included within Grapsoidea. However, their phylogeny included only one cryptochirid species. A later study by *Van der Meij & Schubart (2014)* also used 16s rRNA but contained 10 species, and retrieved the Cryptochiridae as monophyletic and independent from Grapsoidea. Subsequent thoracotreme classification schemes have retained the superfamily

status of the Cryptochiroidea (see overview in *Tsang et al., 2022*), and indicated the need for additional gall crab sequences to elucidate the position of Cryptochiridae in the Thoracotremata, due to weak branch support and uncertainty of tree topology (*Sun et al., 2022*; *Tsang et al., 2022*).

Although Pinnotheridae and Cryptochiridae are considered reciprocally monophyletic, their phylogenetic position within the Thoracotremata—and thus the origin and evolution of thoracotreme symbiosis—is still unresolved (*Sun et al., 2022*; *Tsang et al., 2022*; *Wolfe et al., 2022*; *Kobayashi, Itoh & Nakajima, 2023*). *Sun et al. (2022)* retrieved pea crabs at the basis of a phylogenetic tree of the Thoracotremata, whilst gall crabs clustered, without full branch support, with an Ocypodoidea lineage (Camptandriidae/Xenophthalmidae/Dotillidae). This is in disagreement with the results of *Tsang et al. (2022)* whose phylogenetic arrangement showed both families of symbiotic crabs clustering together—albeit also with inconclusive results—far away from the basal branches in the phylogenetic tree. In a multi-gene and fossil-calibrated phylogenetic reconstruction by *Wolfe et al. (2022)*, Cryptochiridae clustered, again with low support, with Xenograpsidae, whereas *Kobayashi, Itoh & Nakajima (2023)* retrieved, based on mitogenomic data, a close relationship between Cryptochiridae and Grapsidae.

Symbiotic lifestyles in Thoracotremata likely evolved independently on multiple occasions. Aphanodactylidae and Pinnotheridae are not closely related, and Varunidae are retrieved in a distant phylogenetic position from Pinnotheridae and Cryptochiridae (*Sun et al., 2022*; *Tsang et al., 2022*). The question of the evolutionary origin of Cryptochiridae and Pinnotheridae, however, remains open (*Tsang et al., 2022*).

The mitogenomes of most metazoans have high nucleotide substitution rates, low extensive recombination rates and relatively conserved gene content, thus making it an informative molecular signal for phylogenetic reconstruction and adaptive evolution analysis (*Gissi, Iannelli & Pesole, 2008*). Additionally, mitochondrial gene order can provide an extra source of phylogenetic information (*Basso et al., 2017*). Moreover, the life-strategy of a species in response to different environmental pressures may affect the function of mitochondrial genes and exert selective pressure on them. Numerous examples have shown that taxa adapted to inhabiting a specific niche undergo positive selection (*Li et al., 2018*; *Chen et al., 2022*). The range of life strategies, including symbiosis, in the Thoracotremata allow us to study whether there is positive selection in the mitogenomes of these crabs.

So far, a single mitogenome is available for gall crabs, that of the Indo-Pacific species *Hapalocarcinus marsupialis* Stimpson, 1859 s.l. (*Sun et al., 2022*; see *Bähr et al., 2021* for a discussion of the species complex). Cryptochiridae is a peculiar group of diminutive crabs, obligately associated with scleractinian corals. There are currently 55 described species across 21 genera (*DecaNet, 2023*); however, recent studies have uncovered large numbers of undescribed cryptochirid species awaiting formal description (*e.g.*, *Bähr et al., 2021*; *Xu et al., 2022*). Pinnotheridae is a large and diverse family, often associated with invertebrate hosts, containing approximately 330 species in 69 genera, and species new to science are regularly described (*e.g.*, *Ahyong, 2020*; *Ng & Rahayu, 2022*). Currently, mitogenomic data are available for eight symbiotic pinnotherids. Furthermore, mitogenomes of the symbiotic varunids *A. inaequipes* and *T. horvathi* are available for study.

Here we reconstruct a phylogeny of the Thoracotremata using a phylomitogenomic approach based on 75 species (from 14 out of 21 recognised families), including three newly sequenced Cryptochiridae mitogenomes. Based on the inferred phylogeny reconstruction and comparative analysis, we aim to elucidate: (1) the monophyly and phylogenetic position of Cryptochiroidea; (2) the evolution of symbiosis in Thoracotremata; and (3) by means of a test for selective pressure, whether a symbiotic lifestyle results in positive selection for certain protein-coding genes (PCGs) in the mitogenome.

## MATERIALS & METHODS

### Sample collection and mitochondrial genome sequencing

The three gall crab species used in this study were collected from two sites in the Caribbean. *Troglocarcinus corallicola* Verrill, 1908, was sampled from *Orbicella faveolata* (Ellis & Solander, 1786) in Anse à Jacques, Guadeloupe (16°12′29.4″N, 61°25′22.1″W) on April 27, 2021. *Kroppcarcinus siderastreicola* Badaro, Neves, Castro & Johnsson, 2012, collected from *Siderastrea siderea* (Ellis & Solander, 1786) on February 24, 2022, and *Opecarcinus hypostegus* (Shaw & Hopkins, 1977), collected from *Agaricia humilis* Verrill, 1901 on March 14, 2022, were sampled in Piscadera Bay, Curaçao (12°7′18.17″N, 68°58′10.66″W). The crabs were stored in 70% ethanol and transported to the University of Groningen, and from there the entire specimens were sent to the Beijing Genomics Institute (BGI) in Hong Kong for DNA extraction, and paired-end sequencing using the DNBSEQ-G400 platform.

Sampling in Guadeloupe was authorised by the Direction de la Mer de Guadeloupe under Autorisation N°09/2021. Sampling in Curaçao was under collecting permits of the Curaçaoan Government provided to CARMABI (Government reference: 2012/48584).

### Mitochondrial genome assembly and annotation

The raw data were filtered in order to remove adapter, contaminated and low-quality sequences using fastp v.0.23.2 (*Chen et al., 2018b*) with default parameters. The clean data were then assembled with GetOrganelle v.1.7.6.1 (*Jin et al., 2020*) on the Peregrine high performance cluster of the University of Groningen. The assembled mitogenomes were subsequently imported into MITOS1 on the MITOS Web Server (http://mitos.bioinf.uni-leipzig.de/index.py) (*Bernt et al., 2013*) for annotation, and the start and stop codons were confirmed manually using Geneious v.8.1.3. The GCview online service (https://proksee.ca/) was used for the visualisation of the mitogenome maps. The assembled and annotated mitogenomes with gene features were uploaded to GenBank under accession numbers OQ308778 (*K. siderastreicola*), OQ308779 (*O. hypostegus*), and OQ308780 (*T. corallicola*).

### Mitochondrial genome characterisation

After complete annotation of the mitogenome, the nucleotide composition for each species was calculated in MEGA X (*Kumar et al., 2018*) and the composition skew was calculated using the formulas of AT skew $= (A - T)/(A + T)$ and GC skew $= (G - C)/(G + C)$. The Relative Synonymous Code Usage (RSCU) for concatenated PCGs was estimated using the Sequence Manipulation Suite: Codon Usage (https://www.bioinformatics.org/sms2/

codon_usage.html) with the invertebrate mitochondrial genetic code (*Stothard, 2000*) and visualised with the web tool EZcodon (http://ezmito.unisi.it/ezcodon; *Cucini et al., 2021*) combined with the package *ggplot2* (*Wickham, 2016*) in R v.4.2.2 (*R Core Team, 2020*). The transfer RNA (tRNA) genes were identified with MiTFi, implemented in the MITOS web server, using the default settings. The secondary structures of tRNAs were visualised with ViennaRNA Web Services (*Kerpedjiev, Hammer & Hofacker, 2015*). Details of the mitogenome of the cryptochirid *H. marsupialis* s.l. (*Sun et al., 2022*) were missing in the original paper, hence this mitogenome was included in the above-mentioned analysis to allow for better comparison with our newly obtained Cryptochiridae mitogenomes. Using the cytochrome c oxidase subunit I (*cox1*) barcode from the mitogenome by *Sun et al. (2022)* we identified their specimen as *H. marsupialis* HM.08 (Sancia E.T. van der Meij, 2022, unpublished data).

## Phylogenetic analysis

Seventy-two thoracotreme and five heterotreme crabs (as outgroups) for which whole mitogenomes are available were retrieved from GenBank (Table S1). With the addition of the three new gall crab mitogenomes this resulted in a dataset of 80 mitogenomes for phylogenetic inference. No mitogenome data were available for the symbiotic families Parapinnixidae, Tetriasidae and Aphanodactylidae.

The 13 PCGs and two ribosomal RNA genes (rRNAs; rrnS: 12S ribosomal RNA and rrnL: 16S ribosomal RNA) were aligned separately using MAFFT v.7.407 (*Katoh & Standley, 2013*) and subsequently Gblocks v.0.91b (*Talavera & Castresana, 2007*) was applied to remove ambiguously aligned regions using default settings. All PCGs and rRNAs were combined in a concatenated dataset containing 11,193 nucleotides. PartitionFinder 2 (*Lanfear et al., 2017*) was used to detect the best partition scheme, as well as the best-fit nucleotide substitution models for the respective partitions, based on the corrected Akaike Information Criterion (AICc; *Hurvich & Tsai, 1989*).

Maximum Likelihood (ML) and Bayesian Inference (BI) approaches were used for the phylogenetic analyses. The selected partition schemes and best-fit substitution models are available in Table S5. ML was inferred in IQ-TREE v.1.6.8 (*Nguyen et al., 2015*) with 20,000 ultrafast bootstraps (*Minh, Nguyen & Von Haeseler, 2013*), and MrBayes v.3.2.7 (*Ronquist et al., 2012*) was used for the BI analysis; we ran two parallel runs of four chains (one cold and three heated chains) each performing for $10 \times 10^6$ generations, sampling every 1,000 iterations. Consensus trees were constructed in MrBayes with a burnin of 25%. The average standard deviation of split frequencies was 0.03, surpassing the recommended threshold (<0.01) proposed by the software authors as a measure of convergence. The trees were visualised in FigTree v.1.4.2 (http://tree.bio.ed.ac.uk/software/figtree/).

## Comparative analysis of mitochondrial gene order

Besides the five heterotreme crabs as outgroups, the mitogenomes of 75 thoracotreme crabs were annotated by MITOS1 (http://mitos.bioinf.uni-leipzig.de/index.py). The Mitochondrial Gene Order (MGO) of *Xenograpsus testudinatus* Ng, Huang & Ho, 2000, is based on *Ki et al. (2009)* because MITOS1 generated only 20 tRNAs, while it contains 22 tRNAs according to the original paper.

The comparative gene order analysis was conducted using CREx (*Bernt et al., 2007*). In this analysis we compared the MGO of all taxa in our dataset with the ancestral gene order (the most widespread gene order) of the Brachyura (*Basso et al., 2017*). The gene rearrangement scenarios in CREx are based on common intervals, which considers reversals, transpositions, reverse transpositions, and tandem-duplication-random-loss (TDRL) as possible gene rearrangement events, while the control region is excluded from the analysis.

## Analysis of selective pressure

To test for selective pressure on each of the PCGs, the ratio of nonsynonymous (dN) to synonymous (dS) substitution rates ($\omega = dN/dS$) was calculated using the codon-based maximum likelihood (CodeML) application in PAML v.4.7 (*Yang, 2007*). Codon alignments were implemented in the PAL2NAL web server (*Suyama, Torrents & Bork, 2006*). Here we used two models to test for selective pressure: (branch model and branch-site model), and a total of 13 PCGs was computed separately for each of these models. The topology of the ML tree resulting from the phylogenetic analyses, without outgroups and branch lengths, was used for the selective pressure analyses. The downstream analysis excluded unusual $\omega$ ratios, which occurred when the dN or dS values were equal to zero due to limited substitution information from the sequences.

The branch model (free-ratio model) was used to test if the $\omega$ ratios vary between branches in the phylogeny: this model allows an independent $\omega$ ratio for each branch on the phylogenetic tree. Furthermore, the results of $\omega$ ratios in the free-ratio model were classified into two groups: symbionts (Cryptochiridae, Pinnotheridae, varunids *A. inaequipes* and *T. horvathi*) and free-living crabs (all other taxa). The branch-site model (*i.e.,* Model A null *vs* Model A) was applied in this study, which assumes that $\omega$ ratios vary among sites and across the branches of the phylogeny. The branches of three symbiotic lineages were labelled as foreground branches while the rest were set as background branches in the branch-site model in three different topologies only containing one of three symbiotic lineages.

The Wilcoxon rank sum test was used to determine whether the $\omega$ ratios differed significantly between grouped lineages (*i.e.,* symbiotic crabs *vs* free-living crabs) in the free-ratio model. The Likelihood-Ratio Test (LRT) was used to compare branch-site models determining the best fitting model to our data. The parameters setting for LRT followed *Zhang, Nielsen & Yang (2005)*. Then, the Bayes Empirical Bayes (BEB) was used to calculate the posterior probability that each site acted with positive selection under the alternative model in the site models and branch-site models (*Yang, Wong & Nielsen, 2005*). The visualisation was executed using *ggplot2* v3.4.0 (*Wickham, 2016*) in R v4.2.2 (*R Core Team, 2020*).

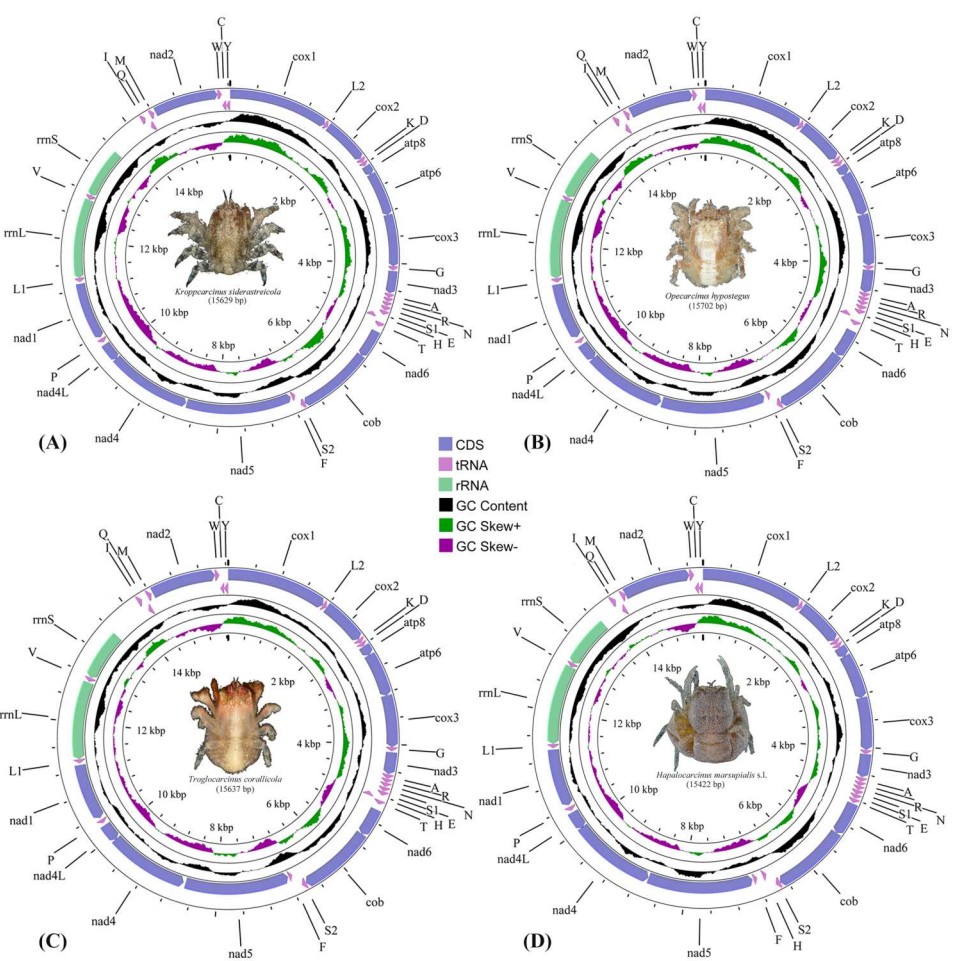

**Figure 1** **The mitochondrial genome maps of (A)** *Kroppcarcinus siderastreicola*; **(B)** *Opecarcinus hypostegus*; **(C)** *Troglocarcinus corallicola*; **and (D)** *Hapalocarcinus marsupialis* **s.l.** The names of the mitochondrial transfer RNA genes can be found in Table S2.

## RESULTS

### Mitochondrial genome

The complete mitogenomes of the cryptochirids *H. marsupialis* s.l., *K. siderastreicola*, *O. hypostegus* and *T. corallicola*, were 15,422 bp, 15,629 bp, 15,702 bp and 15,637 bp in size, respectively. They all contained the usual 13 PCGs, two rRNA genes, 22 tRNA genes and non-coding regions (Fig. 1, Table S2) generally observed in decapod mitogenomes. Twenty-two out of the 37 genes were located on the majority strand (J-strand) and the remaining 15 were located on the minority strand (N-strand).

The nucleotide composition, AT-content and GC-content of the mitogenomes of the 75 thoracotreme crabs are summarised in Table S4. Of the four gall crabs, *K. siderastreicola* had the highest AT-content (74.57%) and *H. marsupialis* s.l. had the lowest value (69.54%).

PCGs comprised 11,144 (*T. corallicola*), 11,149 (*K. siderastreicola*), 11,104 (*O. hypostegus*) and 11,068 (*H. marsupialis* s.l.) codons. Nine PCGs (*cox1*, *cox2*, *atp8*, *atp6*, *cox3*, *nad3*,

*nad6*, *cob* and *nad2*) were located on the J-strand and four (*nad1*, *nad4*, *nad4l* and *nad5*) on the N-strand. The start and stop codons of the four gall crab species are summarised in Table S2.

RSCU and amino acid composition of 13 PCGs for the four gall crabs are shown in Fig. S1. TTA (Leu), ATT (Ile), TTT (Phe) and ATA (Met) were the dominant codons (amino acids) in four gall crabs, and CGC (Arg), CCG (Pro), TCG (Ser) and ACG (Thr) were less commonly used (Table S3).

The majority of tRNAs encoded in the mitogenomes of the four gall crabs showed a "cloverleaf" secondary structure (Figs. S2A–S2D). The dihydroxyuridine (DHU) loop was absent in *tRNA-Ser1* in *T. corallicola*, while *O. hypostegus* and *H. marsupialis* s.l. only had the DHU loop but not the DHU arm. *tRNA-Thr* only had the thymine pseudouracil cytosine (TΨC) arm without the TΨC loop in both *T. corallicola* and *K. siderastreicola*, while the same scenario was found for *tRNA-Asn* and *tRNA-Phe* in *T. corallicola*, tRNA-Asp in *O. hypostegus* and *tRNA-Gly* in *H. marsupialis* s.l. (Figs. S2A–S2D).

## Phylogenetic analysis

ML and BI generated almost identical phylogenetic topologies (Figs. 2, S3); the only noteworthy difference was that *Varuna litterata* (Fabricius, 1798), *V. yui* Hwang & Takeda, 1986 and *Metaplax longipes* Stimpson, 1858 formed a well-supported clade in BI (Fig. S3), whereas *M. longipes* did not cluster with both *Varuna* species in the ML tree (Fig. 2). The superfamilies Ocypodoidea and Grapsoidea were polyphyletic and the two symbiotic superfamilies were retrieved as separate monophyletic clades with strong nodal support (ML = 100, BI = 1). The Pinnotheridae clustered at the base of the phylogenetic tree (ML = 100, BI = 1) in its own independent clade, while the Cryptochiridae lineage was located at a distant phylogenetic position compared to the position of pea crabs. The symbiotic varunids *A. inaequipes* and *T. horvathi* clustered together, with full support, in a clade containing all other Varunidae.

## Comparative analysis of mitochondrial gene order

The comparative analysis of MGO by pairwise comparison with the ancestral gene order of Brachyura detected ten rearrangement patterns in 75 thoracotreme crabs. Within Cryptochiridae, the three Atlantic species *T. corallicola*, *K. siderastreicola* and *O. hypostegus* had the same gene order (pattern eight), which differed from the pattern for the Indo-Pacific species *H. marsupialis* s.l. (pattern seven), where *tRNA-His* changed location and was transposed in between *tRNA-Ser2* and *tRNA-Phe* (Fig. 3). For Pinnotheridae, pattern one, nine and ten were found. *Amusiotheres obtusidentatus* (Dai in Dai, Feng, Song & Chen, 1980), *Pinnotheres pholadis* De Haan, 1835 and *Pinnaxodes major* Ortmann, 1894 shared pattern one, while *Pinnotheres excussus* Dai in Dai, Feng, Song & Chen, 1980, *Arcotheres sinensis* (Shen, 1932), *Arcotheres purpureus* (Alcock, 1900) and *Arcotheres* sp. shared pattern nine; *Indopinnixa haematosticta* (Sakai, 1934) had pattern ten. Pattern three was only detected in *A. inaequipes*, whereas the other symbiotic varunid *T. horvathi* shared pattern two with the remaining Varunidae. As for the free-living crabs, five patterns, including the ancestral gene order of Brachyura, were found across the different families (Figs. 2 and 3).

Peer J

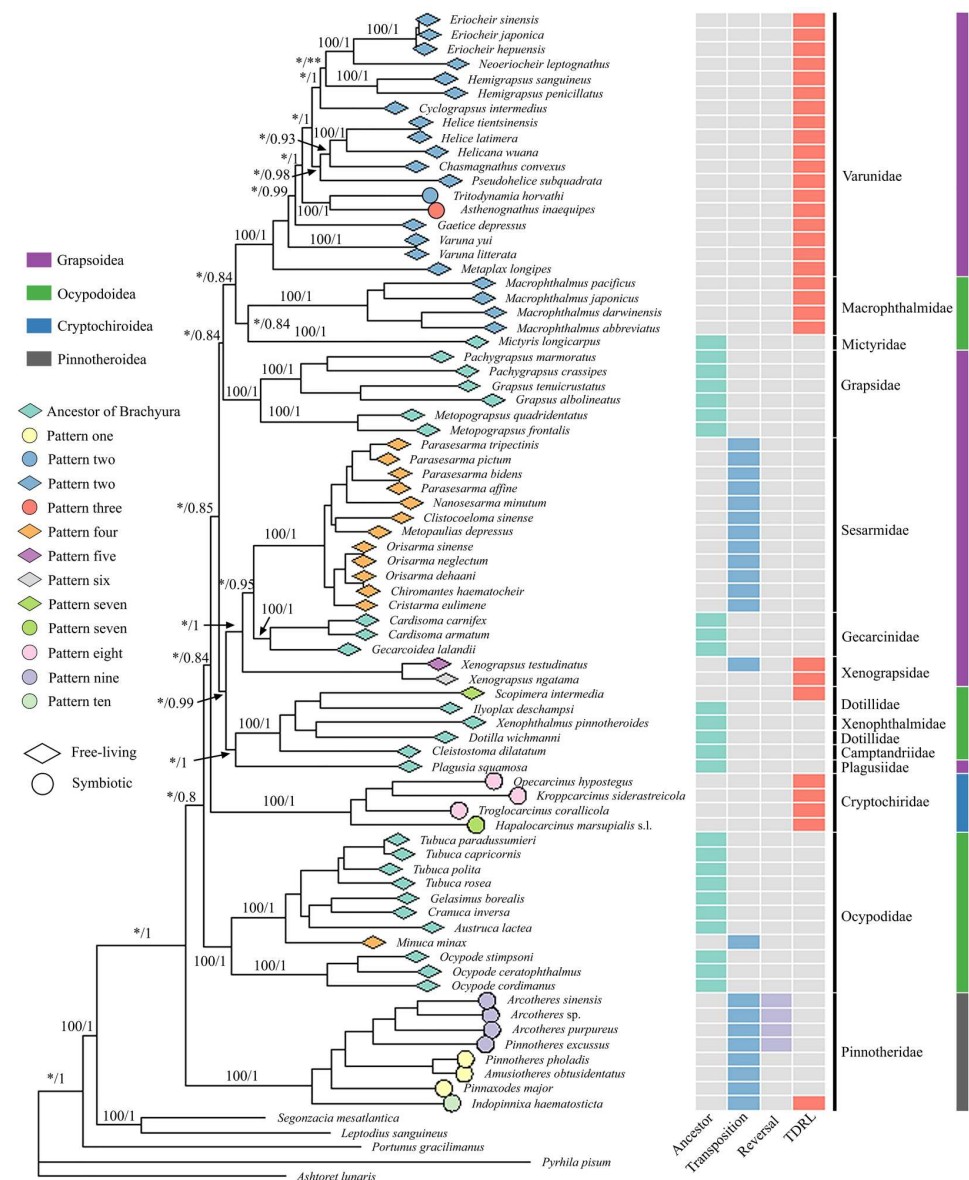

**Figure 2 Phylogenetic tree with mitochondrial gene order information and lifestyle for each species.**
Maximum likelihood (ML) tree based on concatenated genes including 13 PCGs and two rRNA genes.
The branch support values are displayed for the major nodes. Values on the branches refer to Bootstrap
(BP) values of ML (left of the slash) and Posterior Probability (PP) of BI (right of the slash). Stars indicate branch support values under 95 and 80 (one star represents BP and two stars represent PP). Terminal
tips link mitochondrial gene rearrangement patterns (detailed patterns can be found in Fig. 3) detected by
CREx. The leftmost column of an array of coloured boxes refers to the ancestral mitochondrial gene order
of Brachyura, and the remaining columns refer to detailed rearrangement events for each terminal species.
TDRL is the abbreviation for tandem-duplication-random-loss.

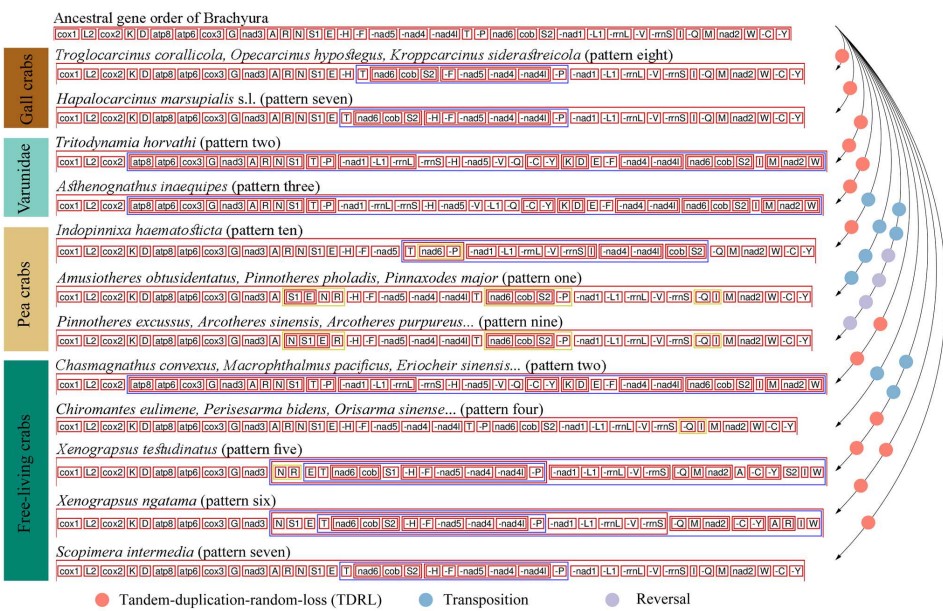

**Figure 3 Mitochondrial gene order patterns detected by CREx for gall crabs, pea crabs, two varunids and free-living thoracotreme crabs.** The solid, coloured circles refer to the number and type of gene rearrangement events. The names of mitochondrial transfer RNA genes can be found in Table S2.

Three out of four types of MGO rearrangement events, including reversals, transpositions and TDRL, were detected in 75 thoracotreme crabs. MGO rearrangement of the four gall crabs was found to be caused by only one TDRL event and *A. inaequipes* and *T. horvathi* exhibited this event twice, while in pea crabs multiple transposition and reversal events were detected. Furthermore, reversal events occurred in three pea crab species in this study (Fig. 3).

## Selection analysis

We used a free-ratio model to calculate selective pressure for terminal branches of the phylogenetic tree. In particular for the free-ratio model, the average value of $\omega$ ratios for nine PCGs was higher for the symbiotic rather than the free-living crabs, with exceptions of the other four genes, *cox2, cox3, nad2* and *nad4l* (Fig. 4; Table S7). The average $\omega$ ratios of *atp6*, *cob* and *nad5* were significantly higher in the symbiotic crabs than in the free-living crabs (Fig. 4, Table S7; Wilcoxon rank sum test, $p = 0.04$, $p = 0.04$ and $p = 0.02$, respectively). Using the branch-site model, two amino acid sites in gall crabs were detected to be under positive selection in *nad6* by BEB (posterior probability $\geq$ 95%), as well as five amino acid sites in *atp8*, one site in *cob* and two in *nad5* with weak positive selection evidence. In pea crabs, genes atp8, *cob*, *nad2*, *nad4*, *nad5* and *nad6*, as well as two genes *atp8* and *nad5* in *Asthenognathus inaequipes* and *Tritodynamia horvathi*, were inferred with weak positive selection evidence (Table S8).

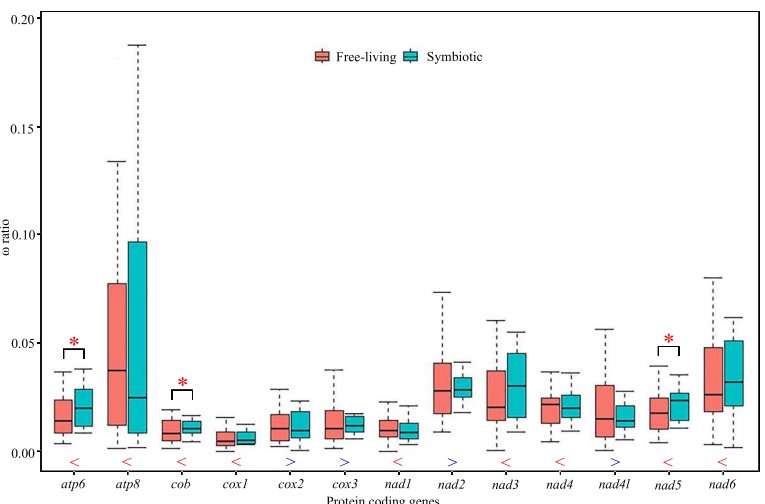

**Figure 4** **Comparison of average *ω* ratios for 13 individual protein-coding genes generated by free-ratio model using CodeML application in PAML between symbiotic and free-living crabs within Thoracotremata.** Red stars indicate that the *ω* ratio in symbiotic species is significantly higher than the one in free-living species ($p < 0.05$). The blue greater-than sign indicates that the average *ω* ratio in free-living lineages is higher than that in symbiotic lineages, whereas the red less-than sign indicates a lower than average *ω* ratio. The outliers are hidden for better visualisation.

## DISCUSSION

### Origin and phylogenetic position of symbiotic crabs

Phylomitogenomics confirmed Cryptochiroidea as an independent monophyletic clade, with robust branch support (ML = 100, BI = 1) within the Thoracotremata (Fig. 2, Fig. S3), deserving of superfamily rank. Earlier studies used a single Cryptochiroidea representative, hence no conclusions about the superfamily status could be drawn based on those trees (*Wetzer, Martin & Boyce, 2009*; *Tsang et al., 2014*; *Tsang et al., 2018*; *Sun et al., 2022*; *Kobayashi, Itoh & Nakajima, 2023*).

A phylomitogenomic study of the Thoracotremata by *Sun et al. (2022)*, based on a similar dataset, retrieved the cryptochirid *H. marsupialis* s.l. as closely related to an Ocypodoidea lineage (Camptandriidae/Xenophthalmidae/Dotillidae), albeit without full branch support. Furthermore, a recent phylogenetic study conducted by *Kobayashi, Itoh & Nakajima (2023)*, based on 82 thoracotreme mitogenomes, retrieved Cryptochiridae as a sister to the Grapsidae clade. Our phylogenetic reconstruction, with three additional cryptochirid mitogenomes, retrieved Cryptochiroidea as a separate lineage with full support. *Tsang et al. (2022)* provided suggestions for taxonomic revisions at superfamilial level, and proposed three new superfamilies in addition to the currently recognised four superfamilies. The main clades in our phylogenetic reconstruction (Fig. 2) are largely in agreement with their multi-marker analyses; however, the lack of mitogenomic data for various families (*e.g.*, Glyptograpsidae, Leptograpsodidae, Aphanodactylidae and Heloeciidae) and the insufficient number of representative species for several families (*e.g.*, Plagusiidae, Mictyridae, Camptandriidae and Xenophthalmidae) do not allow us to
fully compare our results with their proposed new classification. In addition, insufficient representative species within individual families was also problematic in their analyses (*e.g.*, in Leptograpsodidae, Xenograpsidae and Glyptograpsidae). Future studies working on the classification within Thoracotremata need to contain more species given the large number of undescribed species in some families (*Ma et al., 2019*), and the relatively small number of thoracotreme species (out of the more than 1,300 recent Thoracotremata) included in most phylogenetic reconstructions (*Tsang et al., 2014*; *Tsang et al., 2022*; *Sun et al., 2022*; *Kobayashi, Itoh & Nakajima, 2023*).

Symbiosis in Thoracotremata was inferred to have independently evolved at least three times. The phylogenetic trees ML and BI (Fig. 2, Fig. S3) reveal Pinnotheridae and Cryptochiridae as distant, independent clades, eliminating the possibility of a single origin of these symbiotic crabs; a question left unanswered in the study by *Tsang et al. (2022)*. Our result of multiple independent origins of symbiosis in the Thoracotremata is in line with *Van der Meij & Schubart (2014)*, *Wolfe et al. (2022)* and *Kobayashi, Itoh & Nakajima (2023)*. Moreover, a molecular clock approach employed by *Tsang et al. (2014)* revealed that pea crabs (Pinnotheridae) evolved a symbiotic lifestyle before gall crabs. Recently, *Tsang & Naruse (2023)* added two new pea crab families in the Pinnotheroidea: Parapinnixidae and Tetriasidae. Mitogenomic data from these families, as well as the Aphanodactylidae, are currently lacking, but given the distant relationships between the Aphanodactylidae and the Pinotheridae, symbiosis has evolved on more than three occasions in the Thoracotremata with aphanodactylid, pinnotherid, cryptochirid, and varunid lineages. Parallel evolution of symbiosis is not unique for the Thoracotremata. It has also been observed in other taxonomic groups (*e.g.*, Gastropoda (*Goto et al., 2021*); palaemonid shrimps (*Horká et al., 2016*); Copepoda (*Bernot, Boxshall & Crandall, 2021*); *etc.*). The switch from a free-living to a symbiotic lifestyle appears to be relatively common in marine invertebrates (*Horká et al., 2016*).

*Sun et al. (2022)* included *A. inaequipes* and *T. horvathi* (Varunidae) in their dataset, but did not recognise the species as symbiotic. *Astenognathus inaequipes* is symbiotic with holothurians (*Lee, Lee & Ko, 2010*), whereas *T. horvathi* associates with the polychaete *Loimia medusa* (Savigny, 1822) (*Sakai, 1976*; *Otani, Takahashi & Matsuura, 1996*). *Tsang et al. (2022)* considered *T. horvathi* as free-living in their analyses, a decision we disagree with based on *Sakai (1976)* and *Otani, Takahashi & Matsuura (1996)*. Our phylogenetic reconstruction does, however, agree with their assignment of *Tritodynamia* to the Varunidae (*Tsang et al., 2022*; *DecaNet, 2023*).

## Gene arrangement

Gene rearrangements occurred frequently in Thoracotremata (Fig. 3). While metazoan mitogenomes are generally conserved, exceptions exist in Mollusca (*Serb & Lydeard, 2003*), Echinodermata (*Perseke et al., 2008*), Cnidaria (*Kilpert, Held & Podsiadlowski, 2012*), and Decapoda (*Tan et al., 2018*; *Wang et al., 2021*; *Kobayashi, Itoh & Nakajima, 2023*). Interestingly, *Tan et al. (2019)* reported that the occurrence of MGO rearrangements, with pairwise comparisons to the ancestral arthropod ground pattern, is unevenly distributed across decapod infraorders. They reported on four MGO patterns within 37 brachyuran

species, and 13 MGO patterns among 22 anomuran species. However, the diversity of MGO patterns in Brachyura appears to be underestimated. Here we report ten MGO patterns (pattern eight is newly reported here) occurring within the Thoracotremata alone. Our pairwise comparisons were based on the ancestral gene order of Brachyura (*Basso et al., 2017*) and not on the ancestral arthropod ground pattern, which differ by one tRNA gene, however this does not affect the MGO diversity patterns. Comparative analysis confirmed that all symbionts and 59% of the free-living species have undergone variable MGO rearrangement events (Fig. 2). Thoracotreme crabs could potentially exhibit a greater diversity of MGO patterns if more mitogenomes become available. High MGO diversity has also been found in other marine species, including fishes (*Poulsen et al., 2013*), bivalves (*Yang et al., 2019*), echinoderms (*Mu, Liu & Zhang, 2018*; *Galaska et al., 2019*) and worms (*Gonzalez et al., 2021*).

Different MGO patterns were detected in symbiotic gall crabs, pea crabs and varunids. Tandem-duplication-random-loss took place in gall crabs, resulting in two patterns: the Indo-Pacific species *H. marsupialis* s.l. has a unique pattern (pattern seven), and the Atlantic species *K. siderastreicola, O. hypostegus* and *T. corallicola* share the same MGO pattern (pattern eight; Fig. 2). Despite their shared distribution range, the latter three species are not closely related (*Van der Meij & Klaus, 2015*; *Van der Meij & Nieman, 2016*). Transposition, reversal and TDRL events were found in pea crabs, which contributes to them having three MGO patterns (patterns one, nine and ten). Moreover, a unique pattern (three) was found in *A. inaequipes*, while *T. horvathi* shared an MGO (pattern two) with the other Varunidae.

The different mitochondrial gene rearrangement scenarios are perhaps related to their evolutionary history associating with different invertebrate hosts. Gall crabs are obligate symbionts of scleractinian corals, pea crabs associate with a range of invertebrate hosts (but not scleractinian corals), *A. inaequipes* associates with holothurians, whereas *T. horvathi* associates with polychaetes (*Fize & Serène, 1957*; *Sakai, 1976*; *Lee, Lee & Ko, 2010*; *Castro, 2015*; *Palacios Theil, Cuesta & Felder, 2016*; *De Gier & Becker, 2020*). *Sun et al. (2022)* proposed that mitochondrial gene rearrangements may correlate with the specialised lifestyles within the Thoracotremata, however, this does not agree with our results of the MGOs being mostly linked to the various thoracotreme clades (Fig. 2). Moreover, the pea crab *Pinnotheres excussus* has a different MGO pattern from *Pinnotheres pholadis* De Haan, 1835. *Pinnotheres excussus* has been reported as a symbiont of *Gafrarium* bivalves and *P. pholadis* has been reported to live with various bivalve species (*Palacios Theil, Cuesta & Felder, 2016*; *De Gier & Becker, 2020*; Table S1). Both pea crab species are thus associated with bivalve molluscs, which questions a possible correlation between not only specialised lifestyles but also between host association and MGO rearrangements. The reasons contributing to variable MGO patterns observed in different lineages within the Thoracotremata are still unclear.

Identical gene orders shared by relatively distinct organisms were observed in this study. Sharing of the same MGO is extremely rare for distinct taxa with a probability of two mitochondrial genomes sharing identical derived genome organisation being one in 2,664 (*Dowton, Castro & Austin, 2002*; *Babbucci et al., 2014*). Nonetheless, *Babbucci et al. (2014)*

observed the ant *Formica fusca* Linnaeus, 1758 (Hymenoptera) sharing the same MGO with Ditrysia (*e.g.*, *Parnassius bremeri* Bremer, 1864; *Aporia crataegi* (Linnaeus, 1758); *Ochrogaster lunifer* Herrich-Schäffer, 1855), which is the largest clade of Lepidoptera. Here we detected MGO pattern four in 12 grapsoids (*e.g.*, *Parasesarma tripectinis* (Shen, 1940), *Orisarma sinense* (Milne Edwards, 1853), *Cristarma eulimene* (De Man in Weber, 1897; Fig. 2) and one ocypodoid crab (*Minuca minax* (Le Conte, 1855)). Pattern seven of *H. marsupialis* s.l. is shared by *Scopimera intermedia* Balss, 1934, a free-living crab in the Dotillidae (Fig. 2). This result is not in agreement with *Sun et al. (2022)*, who reported *S. intermedia* with the ancestor MGO pattern of Brachyura. The occurrence of distant lineages sharing the same MGO illustrates convergent evolution in these lineages. In general, species sharing the same MGO belong to the same clade/group (with exception of pattern seven), however, the number of species in this study is relatively small when compared to the overall number of extant species (>1,300) in Thoracotremata. Pattern sharing has already been detected in crabs (*Wang et al., 2021*; *Sun et al., 2022*; *Kobayashi, Itoh & Nakajima, 2023*), as well as, for example, in birds (*Mindell, Sorenson & Dimcheff, 1998*), insects (*Babbucci et al., 2014*) and worms (*Weigert et al., 2016*).

Gene order information can be used as an informative character when defining groups at various taxonomic levels (*Babbucci et al., 2014*; *Basso et al., 2017*). MGOs have been shown to effectively supplement phylogenetic analyses to investigate evolutionary systematics in invertebrates (*Boore & Brown, 1998*; *Dowton, Castro & Austin, 2002*; *Basso et al., 2017*). So far gall crabs have two distinct MGO patterns (seven and eight), and pea crabs have three (one, nine and ten; Figs. 2 and 3). The symbiotic varunid *A. inaequipes* is the only crab in our dataset with pattern three. The MGOs, in addition to the phylogenetic analysis, suggests that the symbionts in the Thoracotremata are only distantly related. Combined with their different host usage this is further evidence that these lineages evolved independently. Adaptive evolution (*e.g.*, by transitioning from a free-living to a symbiotic lifestyle) could promote MGO rearrangements as mitogenomes are more likely to be influenced by evolutionary processes than nuclear genes (*Shen et al., 2019*; *Lau et al., 2021*).

## Selection analysis

Positive selection can drive mitochondrial genes to better adapt to a symbiotic lifestyle. Previous studies have assigned positive selection signals detected in animal mitochondrial PCGs to oxygen usage and energy metabolism, as all 13 PCGs in a mitogenome are involved in aerobic metabolism (*Sun et al., 2018*; *Shen et al., 2019*; *Lü et al., 2023*). The branch-site model in this study identified multiple amino acid sites in different genes that experienced positive selection in gall crabs, pea crabs and the symbiotic varunids, although the support for positive selection in the latter two lineages was comparatively weak. These results suggest that adaptive evolution of mitochondrial genes has played a key role in the different energy demand of symbionts. In addition, the free-ratio model detected that the ω ratios for nine PCGs were higher in symbiotic species than in free-living species. Notably, *atp6*, *cob* and *nad5* exhibited significantly higher ratios in symbiotic species than in free-living ones. Perhaps symbiotic species accumulated more non-synonymous mutations, resulting in (likely) advantageous amino acid changes to facilitate adaptation to a symbiotic lifestyle.

In a diverse range of taxa, adaptations to different lifestyles were studied, *e.g.*, flying *vs* flightless in birds and insects, deep *vs* shallow habitats in fish, high *vs* low altitudes in grasshoppers and mud-dwelling *vs* terrestrial habitats in mud shrimps (*Mitterboeck et al., 2017*; *Wang et al., 2017*; *Li et al., 2018*; *Sun et al., 2018*; *Shen et al., 2019*). Our study is the first to compare $\omega$ ratios in symbiotic and free-living species in Thoracotremata, and the importance of mitochondrial genes in shaping the evolution of symbiotic associations.

Detected positive selection scenarios in symbiotic crabs may be caused by differences in reproduction, body size, mobility, or a combination of these (and other) factors, compared with free-living species, and in turn these factors could influence oxygen or energy usage. Cryptochiridae and Pinnotheridae have, for example, a very high reproductive investment when compared to their free-living counterparts (*e.g.*, *Hines, 1992*; *Hartnoll, 2006*; *Bähr et al., 2021*). These symbiotic crabs also live associated with an invertebrate host, resulting in little movement during the adult stage (although this applies mostly to females), but the process of settlement as a megalopa, however, involves swimming or crawling to reach a suitable location on or in the host organism, which might require high energy expenditure. At the same time, certain symbiont species might need to expend energy to overcome host defences, such as the physical barrier on the surface of the host (*Liu, Høeg & Chan, 2016*). The positive selection observed in the mitogenome of symbiotic crabs could potentially be attributed to their unique ecological and physiological characteristics.

## CONCLUSIONS

Cryptochiridae, Pinnotheridae and the varunids *A. inaequipes* and *T. horvathi* were retrieved in three separate lineages in our phylogenetic reconstruction, thus based on our results symbiosis independently evolved at least three times in the Thoracotremata. The recent recognition of two additional pea crab families in the Pinnotheroidea and the elevation of Aphanodactylidae to superfamily level highlights the need for more molecular data to further clarify their positions in the Thoracotremata. However, given the distant relationship between the Aphanodactyloidea and Pinnotheroidea, symbiosis likely evolved at least four times in the Thoracotremata (*Tsang et al., 2022*; *Tsang & Naruse, 2023*). Furthermore, our gene arrangement analysis (Figs. 2 and 3) shows how, in general, the MGO pattern is stable between clades, highlighting the potential usefulness of MGOs in phylogenetic studies. Lastly, our selection analysis revealed strong evidence for positive selection in gall crabs but weak evidence in both pea crabs and symbiotic varunids (*A. inaequipes* and *T. horvathi*). Currently little is known about the exact role and function of these genes, although they are potentially linked to aerobic metabolism. Further research is needed to explore the possible link between mobility, reproduction, body size or other factors that are at play in these positive selection scenarios.

## ACKNOWLEDGEMENTS

We are grateful to Charlotte Dromard (Université des Antilles) and Mark Vermeij (CARMABI) for their support in the field. We also thank the Center for Information Technology (CIT) of the University of Groningen for their support with the Peregrine high

performance computing cluster, and Shane Ahyong (Australian Museum) for providing background information on *Tritodynamia horvathi*. We thank Emma Palacios Theil and two anonymous reviewers for their constructive comments that helped improve an earlier version of this manuscript.

### Funding

Fieldwork by Henrique Bravo in Guadeloupe was funded by TREUB-maatschappij (Society for the Advancement of Research in the Tropics) and Flying Sharks. Fieldwork by Tao Xu in Curaçao was financed by Academy Ecology Fund Grant from the Royal Netherlands Academy of Arts and Sciences (No. KNAWWF/705/ECO202223). The scholarship of Tao Xu was provided by the China Scholarship Council (No. 201907565038). The funders had no role in study design, data collection and analysis, decision to publish, or preparation of the manuscript.

### Grant Disclosures

The following grant information was disclosed by the authors:
TREUB-maatschappij (Society for the Advancement of Research in the Tropics) and Flying Sharks.
Academy Ecology Fund Grant from the Royal Netherlands Academy of Arts and Sciences: KNAWWF/705/ECO202223.
China Scholarship Council: 201907565038.

### Competing Interests

The authors declare there are no competing interests.

### Author Contributions

- Tao Xu performed the experiments, analyzed the data, prepared figures and/or tables, authored or reviewed drafts of the article, and approved the final draft.
- Henrique Bravo performed the experiments, analyzed the data, authored or reviewed drafts of the article, and approved the final draft.
- Sancia E.T. van der Meij conceived and designed the experiments, authored or reviewed drafts of the article, and approved the final draft.

### Field Study Permissions

The following information was supplied relating to field study approvals (*i.e.*, approving body and any reference numbers):

Sampling in Guadeloupe was authorised by the Direction de la Mer de Guadeloupe under Autorisation N°09/2021. Sampling in Curacao was under collecting permits of the Curaçaoan Government provided to CARMABI (Government reference: 2012/48584).

### DNA Deposition

The following information was supplied regarding the deposition of DNA sequences:

The three new assembled and annotated mitogenomes with gene features are available at GenBank: OQ308778 (*Kroppcarcinus siderastreicola*), OQ308779 (*Opecarcinus hypostegus*), and OQ308780 (*Troglocarcinus corallicola*).

## Data Availability
The raw measurements are available in the Supplementary Files.

## Supplemental Information
Supplemental information for this article can be found online at http://dx.doi.org/10.7717/peerj.16217#supplemental-information.

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
