# Peer review of "Phylomitogenomics elucidates the evolution of symbiosis in Thoracotremata (Decapoda: Cryptochiridae, Pinnotheridae, Varunidae)"

_PeerJ, doi:10.7717/peerj.16217_

## Round 0.1 · original submission · Major Revisions

I agree with the reviewers that there are critical concerns regarding the current manuscript that it warrants a major revision. We are particularly concerned about the hypotheses raised by the authors and the analyses conducted. Please kindly provide a point-to-point rebuttal during your revision. In addition, also consider the inclusion of Palacios Theil et al. 2016 Invertebrate Systematics 30(1):1-27 DOI: 10.1071/IS15023, as the supplementary material contains one of the most complete list of hosts reported for pinnotheroids.

Reviewer 1 ·

Basic reporting

I enjoyed reading through the work of Xu et al. The flow and language are clear and smooth, and the framework of the study is solid. The sufficient and up-to-date background provided in the Introduction section facilitates baseline understanding for the readers, especially regarding the debate on the monophyletic status of the superfamilies within Thoracotremata. The addition of three gall crabs' mitogenomes to the currently available one, and using these newly added data to investigate the phylogenetic position of Cryptochiroidea and the evolution of symbiosis in Thoracotremata highlights the novelty of this study. I personally applaud the authors for their high-quality figures!

Experimental design

The overall method is very detailed and well-described. However, did the authors need any permits for sending the entire specimens to Hong Kong?

Line 201: "The mitogenomes of the rest of 70 ..."

Validity of the findings

The bootstrapping performed is sufficient and the clustering of species is clear. The overall discussion is well supported and the authors discussed their findings meticulously. However, may I kindly ask the authors also discuss on the possible difference in the phylogenetic results obtained using ML and BI?

Reviewer 2 ·

Basic reporting

The authors presented a mitogenome phylogeny for the Thoracotremata crab, with a major focus on the phylogenetic position of gall crabs. The authors also performed some analyses on gene order evolution and positive selection on mitogenome for symbiotic lifestyle. I see the value for the phylogenetic work presented here. However, I found there are some major issues in the manuscript concerning the hypotheses to be tested and the analyses conducted. Particularly, I strongly doubt the analyses on positive selection. Firstly, the authors cannot provide some reasonable explanation for how the positive selection on some mitochondrial gene may result in adaptation to symbiotic lifestyle. Secondly, the three symbiotic lineages (gall crabs, pea crabs and Asthenognathus) are polyphyletic that the authors should conduct analyses to test any difference in selection/mutation pattern among the three before analyzing them as a group under “symbiotic” species. Hence, I am a bit skeptical for the results and discussion at this part. I also have some comments for other analyses/discussion listed below that the authors should consider.

Line 109-111 and objective 2 in line 132: It is rather clear that symbiosis have multiple origins in Thoracotremata, regardless of whether Pinnotheridae and Cryptochiridae form a clade or not, because all evidence so far unequivocally show that the symbiotic varunid species and Aphanodactylidae (not mentioned in the manuscript) are distantly related to either Pinnotheridae and Cryptochiridae, and themselves. So, the major question remains is whether Pinnotheridae and Cryptochiridae are closely related (as shown in Tsang et al 2022) or distantly related (as shown in here and Sun et al 2022, of which both adopted mitogenome sequences analyses).

Mitochondrial gene order: the authors only show the gene rearrangements compared to the brachyuran ground pattern in figure 3. However, I think it is important to determine the gene order change among various gene orders (i.e. how a gene order may further evolve into the other as not all new gene orders arise from the ground pattern). This may show some phylogenetic signal and confirm with the topology inferred from the sequence analyses. Furthermore, I am not sure how the authors decide which is “the ancestral pattern for Brachyura”. There is a widely acknowledged/adopted pancrustacean ground pattern, but I don’t personally know a universally accepted “brachyuran ground pattern”.

Figure 3: it is not necessary to show species with the same mitogenome gene order. Lastly, it looks odd to me that the authors also refer Asthenognathus as “one varunid”. This may confuse the readers.

Figure 1: only 3 gall species were sequenced in the present study. It looks a bit odd to me that the authors also show the mitogenome feature for the gall crab sequenced in other study given the topic is “symbiosis” rather than “gall crabs”.

Figure 2: the box in the figure is not really a heat map that should contain “temperature” indicating levels.

Experimental design

see above

Validity of the findings

see above

Additional comments

nil

·

Basic reporting

- The article is well written, in a correct English for the most part. I have made some suggestions for article usage, or punctuation, although some might be considered a case of subjective taste, rather than grammatically correctness.

- The literature references added show an appropriate level of field background.
Nevertheless, I am missing an important reference: Kobayashi et al. 2023 showed a few months ago a very similar phylogenetic study based also on the mitogenomes available at that time for thoracotremes.
Their results should be discussed and compared to those obtained here. In addition, Kobayashi et al. 2023 obtained the mitogenome for an additional pinnotherid species (Indopinnixa heamatosticta). This species belongs to a different subfamily than all the other pinnotherids included in the analyses. It could be interesting to compare it to them.

The classification for two of the four superfamilies discussed here has been updated recently and the assignments for the families used here should be updated.

There are also some major generalisations and extrapolations made here that are, in my opinion, not justified. It is assumed that all symbiotic species studied here are obligate symbionts, when that has never been explored properly for many of the species treated. In addition, the authors assume that all members of the families Cryptochiroidea and Pinnotheroidea are obligate symbionts, although they do acknowledge at some point that the sample number is low and future additional sampling is necessary. For example they include six pinnotherid species in their analyses, when there are currently about 300 species in the superfamily.

- The number, structure and clarity of the figures is convenient.
In some cases I have wondered, however, why some of the results were added as supplementary material, rather than in the results section.
I think that a table indicating the species, associated hosts, locations for the newly obtained cryptochiroid species would add clarity.

Please see comments in attached pdf file for further details.

Experimental design

- the paper meets the aims and scope of the journal (Biological Sciences).
- the research questions are well defined, and the addressed gaps are identified.
- the investigation is performed to an appropriate technical and ethical standard, although in my opinion further points should be added to the discussion (see point 1 and comments in attached pdf file for further details).
- the methods are properly described and information is provided to allow replication. However, I miss proper argumentation for the value of mitogenomes as a molecular tool for building phylogenies, when compared to other methods. Large phylogenies have been built recently for Brachyurans based on multi-genes molecular data, transcriptomes or phylogenomics that in some cases provide better resolution. In addition, the number of available samples might be higher than for mitogenomes. It seems to me that it might provide better results to obtain the same kind of data for the four cryptochiroid species studied here, and add it to those larger phylogenies. For example the phylogeny built in Wolfe et al. 2022 already includes three cyrptochiroid and eight pinnotherid species. Wolfe et al. 2022 is already cited in the paper reviewed.

Validity of the findings

- the validity of the findings is meaningful for the points addressed. However, I find that some questions are left unanswered.
For example, the phylogeny built shows that the analysed species of the superfamilies Pinnotheroidea and Cryptochiroidea form distinct monophyletic clades. This is presented as an indication that symbiosis evolved independently in those two taxa, and previous studies addressing this same questions are compared with this. However, the phylogeny built might have some flaws.The clades observed for many of the other taxa included in the analyses do not agree with other phylogenies built for similar groups, also based on molecular data. This incongruences are not addressed at all in the discussion.

- Most conclusions are well stated. Nevertheless, there are sometimes some assumptions made that seem based on hypothetical theories, rather than in previous observations or confirmed data.
(please see comments in attached pdf file for further details).

Additional comments

Thank you for this paper.
Please see comments in attached pdf file for further details.

I am clicking the "major revisions". I feel however, that the suggestions made should not represent a large amount of work. Nevertheless, I think that some recommendations are too important to be considered "minor". The main ones are:

- update the superfamily assignments according to the new taxonomical classification
- discuss the incongruences observed when comparing this phylogeny with other phylogenies (for ex. Tsang et al. 2022 or Wolfe et al. 2022)
- avoid generalisations of the type of symbiosis displayed by these taxa
- compare findings with those in Kobayashi et al. 2023
If possible, add Indopinnixa to the analysis.

---

## Round 0.2 · Minor Revisions

It's evident from the reviewers' feedback that the revised manuscript has made substantial enhancements in terms of content and the overall structure. Nonetheless, there are still some minor revisions that are necessary. Specifically, the utilization of the term "obligate symbionts" should be carefully re-evaluated.

Reviewer 1 ·

Basic reporting

The revised manuscript shows great improvement in terms of clarity and content, especially the Introduction section. However, I would recommend the authors to get a final round of proofreading before publication. In addition, below are some minor comments that I think might be beneficial for improving the quality of the manuscript:
1. Line 61-62: suggest to remove the two species names to show consistency of just highlighting the family names.
2. Line 63: this is the first mention of ‘pea crab’ instead of ‘gall crab’. What are ‘pea crab families’? Readers might not be able to understand.
3. Line 392: I think ‘only one’ mitogenome sounds better than ‘a single mitogenome’
4. Line 1689: do you mean ‘were’ retrieved?
5. Line 1694: Start a new sentence with ‘However, given the..’

Experimental design

-

Validity of the findings

-

Additional comments

-

Reviewer 2 ·

Basic reporting

The revised version of the manuscript is much improved and I think it is up to the standard of acceptance. I only have a few minor edit and some updated references that the authors should consider.

Introduction: maybe good to mention the two new families recognized by Tsang & Naruse in the introduction so reader know better about pea crab systematics, especially the authors used Pinnotheridae and Pinnotheroidea interchangably throughout. It may eas the reader what is inside Pinnotheroidea. (the paper was acknowledged in discussion)

Poore & Shane 2023 place symbiotic Aphanodactylidae into a new superfamily Aphanodactyloidea with explanation.The authors may update this piece of information in the text.

Line 98: reciprocal monophyletic?

Figure 2 key: "obligate symbiotic" or "symbiotic", to be consistent wtih fig 4 key and the text.

Figure 3: "Symbiotic" Varunidae?

Experimental design

no comment

Validity of the findings

no comment

·

Basic reporting

This paper is written for the most part in a clear and unambiguous way, with a professional English usage.
The references are appropriate and reflect sufficient background research and provide adequate context.
The article structure is also adequate. The figures are clear and illustrate and summarize the results with a visually attractive and intuitive style.
I could not find the links to some of the supplementary tables cited, however I am assuming they are appropriate as well, judging by the tables to which I do have access.
The results add interesting pieces of information, especially for Cryptochiroidea, they multiply the number of available mitogenomes by four. In addition, interesting new ways of analysing these new and already available data are presented.

Experimental design

The experimental design and the methods are carefully explained to ensure replicability. The research question is well defined, addressed and discussed, properly identifying, and tackling knowledge gaps.

Validity of the findings

The findings appear to be valid, obtained with statistically sound methods and the conclusions address the stated questions, discussed in comparisons with related relevant studies. Gaps and potential future prospectives are identified.

Additional comments

The revision has improved the paper, and tackled all suggestions made by the reviewers.
Thank you very much for taking the time to address each one of the comments made previously. It has been enlighting and very interesting to read.
See minor suggestions and comments in the attached document.


Besides those comments, I have only one main issue with this version: The generalization of the studied species, specially pinnotherids, as OBLIGATE symbionts.
It might be due to us using different definitions for the concept.

For me an obligate symbiotic species is the one that would not survive without its hosts.

For pinnotherids (and probably for Aphanodacylidae and varunids as well, but I haven’t done that research):
- At an individual level that is only the case for ovigerous females. And only for those that are endosymbiotic, mostly in molluscs, but also in sea cucumbers and tunicates.
Pinnotherid larvae, megalopa, pre-copulatory females, and males, have little problem surviving without a host. Or they have the same problems that any small free-living crab would also face.
Ectosymbiotic pinnotherids (and those associated with worm tubes) can survive without a host. There are studies that show they can switch hosts within a life time (for ex. De Bruyn et al. 2010), what suggest they might be FUNCTIONAL symbiotic rather than obligate.
- At the species level, they would not survive if the reproductive strategies would depend on the hosts. That can be true for some pinnotherids, namely for those pinnotherids where the female settles down within a host before fertilization, and the males search for them, probably using cues related to that specific host type. But there are other pinnotherids that build swarms in the water, fecundation takes place, and then it is just the female the one looking for an appropriate shelter. Unfortunately, the reproductive strategies for the vast majority of pinnotherids are still unknown. Therefore, in my opinion, it is premature to assume that they all are obligate rather than functional symbionts.

In their response the authors indicate that now in table S1 they have added information about the hosts for the symbiotic species included in the study. I don’t think that the reported hosts for those species is evidence of them being obligate symbionts. Only evidence of them being more of less host-specific. We should also keep in mind that for many small crabs the hosts remain very frequently unreported.

I acknowledge that Castro (2015) did write that most pinnotherids are obligate symbionts. I do admire and really appreciate Pedro. However, he too might have resorted to a wide generalisation regarding this. The cited reference is after all a review on all brachyuran symbionts. Or he might have applied a less strict definition of “obligate” than mine.
Besides, and although it is always nice if you can back up your findings with previous research, the fact that somebody wrote something down, does not make it an absolute truth. Peter Casto himself would most likely agree with this.

Therefore, I still suggest to NOT describe the included species as “obligate symbionts” but simply as “symbionts”. I don’t think the findings or the discussion presented would lose any of their validity if addressing symbiosis, rather than obligate symbiosis.

OR

If the authors think it is important to classify the studied symbiotic groups as obligate, I would suggest to include the definition of obligate symbiosis taken into account.

---

## Round 0.3 · accepted · Accept

I applaud the authors for following through with the comments and suggestions of the reviewers. I believe the manuscript is now ready for publication.